# Planning with Object Creation

**Primary Keywords:** *None*

## Abstract

Classical planning problems are defined using some specification language, such as PDDL. The domain expert defines action schemas, objects, the initial state, and the goal. One key aspect of PDDL is that the set of objects cannot be modified during plan execution. While this is fine in many domains, sometimes it makes the modeling much more complicated. This not only impacts the performance of the planners, but it also requires the domain expert to bound the number of required objects beforehand, which might be an intractable problem by itself. Here, we introduce an extension to the classical planning formalism, where action effects can create and remove objects. This problem is semi-decidable, but it becomes decidable if we can bound the number of objects at any given state, even though the state-space is still infinite. On the practical side, we extend the Powerlifted planning system to support this PDDL extension. Our results show that Powerlifted does not lose efficiency by supporting this extension while allowing for easier PDDL models.

## Introduction

Bob, a former classical planning researcher, is opening a new logistics company. Real life, however, is not so simple. He first needs to decide how many trucks he needs to buy. Buying many trucks is not an issue – Bob became very rich working with classical planning – but he still wants to minimize his expenses. He decides to tackle this problem using classical planning. Bob encodes the map of his city and the delivery locations in PDDL (McDermott 2000; Haslum et al. 2019). He then declares a bunch of truck *objects* in advance, and cleverly encodes his actions to balance the costs between buying a new truck and doing more deliveries with a same truck. But how does he know how many trucks to declare in advance? Does he compute an estimate upper bound or does he overshoot this bound? Computing it seems as if he is solving the problem *himself*, so he goes with a rough estimate of 10 trucks. But does the optimal solution only require 10 trucks? What if it requires much more? Bob gets worried, and increases the number of truck objects to $1,000$.

He finally formalizes his problem and runs it on some classical planners. All planners take months to solve his task. At the end, the optimal plan uses 11 trucks. Even if $99.8\%$ of the objects were irrelevant, they still impacted the performance of the planners (Fuentetaja and de la Rosa 2016;

Silver et al. 2021). Bob ends up frustrated with the whole procedure. If only there was a native way to let planners introduce more objects as they plan.

In this paper, we introduce a novel way of dealing with problems where the objects are not all known upfront. Instead of preemptively declaring all objects in the definition of the task, new objects can be *created* and also *removed* via action effects. There are three clear benefits to this extension: (i) it makes the encoding simpler and more natural for several domains (e.g., Long and Fox 2003), (ii) it reduces the amount of expert knowledge need in the domain encoding (e.g., Petrov and Muise 2023), and (iii) it might improve performance of planners by reducing state size and number of unnecessary objects (e.g., Fuentetaja and de la Rosa 2016).

On the theoretical side, classical planning with object creation is semi-decidable. In other words, if a plan exists we are guaranteed to find it. However, no algorithm will recognize, in general, if a task is unsolvable. If we talk about *optimal* planning though, the problem becomes decidable. Furthermore, for tasks with *bounded states* we show decidability, even though the state-space is still infinite.

On the practical side, we introduce a PDDL extension that allows for object creation/removal in the effect of actions. More importantly, we extend the state-of-the-art lifted planner Powerlifted (Corrêa et al. 2020) to deal with our formalism. Powerlifted *has better performance* when using the extended PDDL, in comparison with the original PDDL encodings where all objects are declared beforehand and object creation/removal is simulated with auxiliary predicates. Overall, this is a first implementation showing object creation is feasible and useful for many domains.

## First-Order Logic

We consider *first-order languages* $\mathcal{L} = \langle \mathcal{V}, \mathcal{P} \rangle$, where $\mathcal{V}$ is a finite set of *variables* and $\mathcal{P}$ is a finite set of *predicate symbols*. Each predicate symbol $P \in \mathcal{P}$ has an associated *arity* $ar(P) \in \mathbb{N}_0$. We restrict our definitions to languages without constant or function symbols.[1] Thus, a *term* is a variable. An *atom* $P(t_1, \ldots, t_{ar(P)})$ is composed of a predicate symbol $P \in \mathcal{P}$ and a tuple of terms $\langle t_1, \ldots, t_{ar(P)} \rangle$. We assume

---

[1]We will discuss later how constant symbols can be emulated, so not allowing them syntactically is no loss of generality.

familiarity with common concepts from first-order logic.

An *interpretation* over a first-order language $\mathcal{L}$ is a tuple $\mathcal{I} = \langle \mathcal{U}^{\mathcal{I}}, \{P^{\mathcal{I}}\}_{P \in \mathcal{P}} \rangle$ consisting of

- a finite[2] set $\mathcal{U}^{\mathcal{I}}$ of *objects* called the *universe*, and
- for each predicate symbol $P \in \mathcal{P}$, its *interpretation* $P^{\mathcal{I}} \subseteq (\mathcal{U}^{\mathcal{I}})^{ar(P)}$.

A *variable assignment* is a function $\sigma$ (partial or total) that maps the variables $\mathcal{V}$ to objects. For a variable $v$ and object $t$, we write $\sigma[v/t]$ for the assignment that agrees with $\sigma$ on variables other than $v$ and maps $v$ to $t$. With some abuse of notation, we extend $\sigma$ to atoms, e.g., $\sigma(P(v,w)) := P(\sigma(v), \sigma(w))$.[3]

As usual, we write $\mathcal{I}, \sigma \models \varphi$ to denote that a formula is true for a given interpretation and assignment. This is defined as usual in first-order logic with one extension. Under regular first-order logic conventions, variable assignments are always total, and we must have $\sigma(\mathcal{V}) \subseteq \mathcal{U}^{\mathcal{I}}$ for the notation $s, \sigma \models \varphi$ to be legal. In this paper, we do allow assignments to map variables to objects that are not part of the universe of $\mathcal{I}$, but atoms involving such variables will always be false, i.e., $\mathcal{I}, \sigma \not\models P(v_1, \ldots, v_n)$ if for some $i$ we have that $\sigma(v_i) \notin \mathcal{U}^{\mathcal{I}}$. This does not require a modification to the standard semantics of atoms (if $\sigma(\mathcal{V}) \subseteq \mathcal{U}^{\mathcal{I}}$ then define $\mathcal{I}, \sigma \models P(v_1, \ldots, v_n)$ iff $\langle \sigma(v_1), \ldots, \sigma(v_n) \rangle \in P^{\mathcal{I}}$), but is rather an extension of which notations are considered legal, i.e., we extend the notation $\mathcal{I}, \sigma \models \varphi$ to cases where $\sigma(\mathcal{V}) \not\subseteq \mathcal{U}^{\mathcal{I}}$. This extension will prove convenient for defining the semantics of object-creation effects in the following section (in particular, for conditional effects).

## Planning Formalism

A *planning task with object creation* is a tuple $\Pi = \langle \mathcal{L}, s_0, G, \mathcal{A} \rangle$, where $\mathcal{L}$ is a first-order language; $s_0$ is the *initial state*, which is an interpretation over $\mathcal{L}$; $G$ is the *goal*, which is a closed formula over $\mathcal{L}$; $\mathcal{A}$ is a finite set of *action schemas*, defined below. In the context of planning tasks, interpretations are known as *states*, and it is customary to denote them by $s$.

**Example 1.** *We use the logistics scenario from the Introduction as a running example. Variables are denoted using upper case letters ($T, L$, etc.), and objects as lower case letters ($c_1, t$, etc.). The predicate symbols are $at, in, connected$ (binary) and $headquarters$ (unary). The initial state $s_0$ has:*

$$\mathcal{U}^{s_0} := \{p_1, c_1, c_2\}$$
$$\mathcal{P}^{s_0} := \{connected(c_1, c_2), connected(c_2, c_1),$$
$$at(p_1, c_1), headquarters(c_1)\}.$$

*(We write $\mathcal{P}^{s_0}$ as a set of atoms here to simplify notation.)*

*We define the goal $G := at(p_1, c_2)$.*

*Intuitively, our task has a package $p_1$ initially located at city $c_1$. There are two cities, $c_1$ and $c_2$, which are connected.*

*The city $c_1$ is the headquarters of the company. The goal is to move $p_1$ from $c_1$ to $c_2$.*

*The action schemas are defined in the next example.*

An action schema $A = \langle pre(A), eff(A) \rangle$ consists of a *precondition* $pre(A)$ and an *effect* $eff(A)$. The precondition $pre(A)$ is a first-order formula. Effects are defined inductively as follows:

- every atom $a$ and its negation $\neg a$ are effects, called *simple effects*;
- if $e_1, e_2$ are effects, then $(e_1 \wedge e_2)$ is a *conjunctive effect*;
- if $\varphi$ is a first-order formula and $e$ is an effect, then $(\varphi \triangleright e)$ is a *conditional effect*;
- if $v \in \mathcal{V}$ is a variable and $e$ is an effect, then $\forall v : e$ is a *universal effect*;
- if $v \in \mathcal{V}$ is a variable and $e$ is an effect, then $\oplus v : e$ is an *object-creation effect*;
- if $v \in \mathcal{V}$ is a variable, then $\ominus v$ is an *object-removal effect*.

As usual we may drop parentheses when there is no resulting ambiguity, e.g., we may write $e_1 \wedge e_2 \wedge e_3$ instead of $((e_1 \wedge e_2) \wedge e_3)$. Apart from object-creation and object-removal effects, these definitions follow the PDDL formalism (McDermott et al. 1998; Haslum et al. 2019), except that we use logic notation. Intuitively, an object-creation effect creates a new object and binds it to a variable $v$ within the effect. An object-removal effect removes an object in the resulting state.

**Example 2.** *Define the following action schemas:*

$$pre(buy) := headquarters(L)$$
$$eff(buy) := \oplus T : at(T, L)$$
$$pre(sell\text{-}all) := \neg headquarters(L)$$
$$eff(sell\text{-}all) := \forall T : at(T, L) \triangleright \ominus T$$
$$pre(move) := at(T, L)$$
$$eff(move) := \neg at(T, L) \wedge at(T, M)$$

*Thus, action buy says "if the $L$ is the headquarters, then add a new truck to $L$"; sell-all says "if location $L$ is not the headquarters, sell all the trucks at $L$"; and move says "move a truck $T$ from location $L$ to $M$".*

We extend the definition of *free variables* from formulas to effects:

- for a simple effect $e$, $free(e)$ is the set of variables appearing in $e$;
- $free(e_1 \wedge e_2) = free(e_1) \cup free(e_2)$;
- $free(\varphi \triangleright e) = free(\varphi) \cup free(e)$;
- $free(\forall v : e) = free(e) \setminus \{v\}$;
- $free(\oplus v : e) = free(e) \setminus \{v\}$.
- $free(\ominus v) = \{v\}$.

We also extend the definition to action schemas as $free(A) = free(pre(A)) \cup free(eff(A))$. Free variables of action schemas correspond to *parameters* in PDDL.

Next, we want to define semantics for action application, for which purpose we introduce the *changes* function. This function maps a state, variable assignment and effect to a tuple of *new objects*, *removed objects*, *added atoms*, and *deleted atoms*. We define $\text{changes}(s, \sigma, \textit{eff})$ inductively:

---

[2]Interpretations in the context of classical planning are always finite.

[3]Note that for atoms $a = P(v_1, \ldots, v_n)$, $\sigma(a)$ is not formally an atom because $\sigma(v_i)$ is not a term. We can think of $\sigma(a)$ as the pair $\langle P, \langle \sigma(v_1), \ldots, \sigma(v_n) \rangle \rangle$ consisting of a predicate symbol and a tuple of objects.

- if *eff* is a positive simple effect $a$, then
$$\text{changes}(s, \sigma, \textit{eff}) = \langle \emptyset, \emptyset, \{\sigma(a)\}, \emptyset \rangle;$$
- if *eff* is a negative simple effect $\neg a$, then
$$\text{changes}(s, \sigma, \textit{eff}) = \langle \emptyset, \emptyset, \emptyset, \{\sigma(a)\} \rangle;$$
- if $\textit{eff} = (e_1 \wedge e_2)$, then
$$\text{changes}(s, \sigma, \textit{eff}) = \langle \text{New}_1 \cup \text{New}_2, \text{Rem}_1 \cup \text{Rem}_2,$$
$$\text{Add}_1 \cup \text{Add}_2, \text{Del}_1 \cup \text{Del}_2 \rangle,$$
where $\langle \text{New}_i, \text{Rem}_i, \text{Add}_i, \text{Del}_i \rangle = \text{changes}(s, \sigma, e_i)$ for $e \in \{1, 2\}$.
- if $\textit{eff} = (\varphi \triangleright e)$ then[4]
$$\text{changes}(s, \sigma, \textit{eff}) = \begin{cases} \text{changes}(s, \sigma, e) & \text{if } s, \sigma \models \varphi, \\ \langle \emptyset, \emptyset, \emptyset, \emptyset \rangle & \text{otherwise}; \end{cases}$$
- if $\textit{eff} = \forall v : e$, then
$$\text{changes}(s, \sigma, \textit{eff}) = \langle \bigcup_{u \in \mathcal{U}^s} \text{New}_u, \bigcup_{u \in \mathcal{U}^s} \text{Rem}_u,$$
$$\bigcup_{u \in \mathcal{U}^s} \text{Add}_u, \bigcup_{u \in \mathcal{U}^s} \text{Del}_u \rangle,$$
where
$$\langle \text{New}_u, \text{Rem}_u, \text{Add}_u, \text{Del}_u \rangle = \text{changes}(s, \sigma[v/u], e)$$
for all $u \in \mathcal{U}^s$.
- if $\textit{eff} = \oplus v : e$, then
$$\text{changes}(s, \sigma, \textit{eff}) = \langle \text{New}' \cup \{\textit{new}\}, \text{Rem}', \text{Add}', \text{Del}' \rangle,$$
where *new* is a unique fresh object,[5] and $\langle \text{New}', \text{Rem}', \text{Add}', \text{Del}' \rangle = \text{changes}(s, \sigma[v/\textit{new}], e)$.
- if $\textit{eff} = \ominus v$, then
$$\text{changes}(s, \sigma, \textit{eff}) = \langle \emptyset, \{\sigma(v)\}, \emptyset, \emptyset \rangle,$$

**Example 3.** *Let $\sigma$ be an assignment with $\sigma(L) = c_1$. Then*
$$\text{changes}(s_0, \sigma, \textit{eff}(buy)) = \langle \{t\}, \emptyset, \{at(t, c_1)\}, \emptyset \rangle$$
*where $t$ is a fresh object not in $\mathcal{U}^{s_0}$.*

We are now ready to complete the definition of action semantics. A *ground action* $A^{s,\sigma}$ is given by an action schema $A$, a state $s$, and a variable assignment $\sigma$ that maps the free variables of $A$ to elements of $\mathcal{U}^s$ (its definition for other variables does not matter). We say that the ground action $A^{s,\sigma}$ is *applicable* in a state $s$ if $s, \sigma \models \textit{pre}(A)$. The *successor state* $\textit{succ}(s, \sigma, A)$ under $A^{s,\sigma}$ is the state defined as follows. Let $\text{changes}(s, \sigma, \textit{eff}(A)) = \langle \text{New}, \text{Rem}, \text{Add}, \text{Del} \rangle$. For each $P \in \mathcal{P}$, let
$$\text{Add}^P = \{\langle t_1, \ldots, t_n \rangle \mid \langle P, \langle t_1, \ldots, t_n \rangle \rangle \in \text{Add}\}$$
$$\text{Del}^P = \{\langle t_1, \ldots, t_n \rangle \mid \langle P, \langle t_1, \ldots, t_n \rangle \rangle \in \text{Del}\}$$
Then $\textit{succ}(s, \sigma, A) = \langle \mathcal{U}', (P')_{P \in \mathcal{P}} \rangle$, where
$$\mathcal{U}' = (\mathcal{U}^s \setminus \text{Rem}) \cup \text{New}$$
$$P' = \{t \in (P^s \setminus \text{Del}^P) \cup \text{Add}^P \mid t \in \mathcal{U}'\}.$$

---

[4]This is the only place where we use our notation that $s, \sigma \not\models P(v_1, \cdots, v_n)$ if $\sigma(v_i) \notin \mathcal{U}^s$ for some $i$.

[5]A fresh object is an object that is not in $\mathcal{U}^s$. We explain later a simple way on how to choose *new*. The supplementary material also contains a more thorough discussion on this topic, and how to guarantee uniqueness.

**Example 4.** *Using $\sigma$ from the previous example, the successor state $s_1 = \textit{succ}(s, \sigma, buy)$ is:*
$$\mathcal{U}^{s_1} := \{p_1, c_1, c_2, t\}$$
$$\mathcal{P}^{s_1} := \{\textit{connected}(c_1, c_2), \textit{connected}(c_2, c_1),$$
$$\textit{at}(p_1, c_1), \textit{at}(t, c_1), \textit{headquarters}(c_1)\}.$$

A *plan* is a finite sequence of ground actions $A_0^{s_0,\sigma_0}, \ldots, A_n^{s_n,\sigma_n}$ such that, $A^{s_i,\sigma_i}$ is applicable in state $s_i$ for $0 \leq i \leq n$, and $s_{i+1} = \textit{succ}(s_i, \sigma_i, A_i)$ for $0 \leq i < n$, and $\textit{succ}(s_n, \sigma_n, A_n) \models G$.

**Remark 1.** *Our logical language does not have constant symbols. We decided so because we do not want to commit to what happens if an object named by a constant symbol is removed (e.g., should the constant symbol interpret a different object?). Instead, we observe that constants can be emulated in our definition of planning with object creation.*

*Suppose $\mathcal{C}$ is a finite set of constant symbols that may appear in the goal and the action schemas. For each $c_i \in \mathcal{C}$, introduce a unary predicate $C_i$. In the initial state, ensure that the interpretation of each $C_i$ predicate consists of a single object. We replace an action $A$ by an action $A'$ as follows. Let $\{c_1, \ldots, c_m\} \subseteq \mathcal{C}$ be the set of constants symbol occurring in $A$. We define $\textit{pre}(A')$ as*
$$\textit{pre}(A') = \exists x_1. \ldots \exists x_m. \left( \bigwedge C_i(x_i) \right) \wedge \textit{pre}(A)[c_i/x_i]$$
*where $\textit{pre}(A)[c_i/x_i]$ indicates the original precondition but replacing each constant $c_i$ with the respective variable $x_i$. Intuitively, this ensures that none of the objects relevant to $A$ (those interpreted by constants that are mentioned in $A$) have been removed. If they have have, however, the action becomes inapplicable. We also define $\textit{eff}(A') = \textit{eff}(A)[c_i/x_i]$. Last, the goal $G$ is modified in a similar fashion, by quantified its constants on the new predicates.*

*This construction encodes that, whenever an object associated to constant symbol is removed, all action schemas referring to this constant are inapplicable. It also encodes that once an object mentioned in the goal is removed, the goal is no longer reachable.*

We introduce two natural decision problems:

**Definition 1** (OBJCREATION-PLANEX)**.** *Given a planning task with object creation $\Pi$, is there a plan for $\Pi$?*

**Definition 2** (OBJCREATION-PLANLEN)**.** *Given a planning task with object creation $\Pi$ and $k \in \mathbb{N}$, is there a plan for $\Pi$ with length at most $k$?*

Classical planning problems can be solved in many different ways, e.g., satisfiability (Kautz and Selman 1992), heuristic search (Bonet and Geffner 2001). In this work, we will extend the planning-as-heuristic-search paradigm to planning with object creation. We assume that the reader has familiarity with search algorithms and their terminology.

## PDDL Extension

We extended the PDDL syntax with the keywords : new and : remove. The first one allows for the creation of objects, and the second for their removal. Their syntax is as follows:

$$(\text{: new } (?v1 \ldots ?vN) (\text{eff}))$$
$$(\text{: remove } (?v1 \ldots ?vN))$$

where v1, ..., vN are variables and eff is an effect. In contrast to the logic formalism above, the PDDL syntax allows to create or remove many objects at once. This can be easily reproduced within our formalism. For object creation, the PDDL encoding is equivalent to $\oplus v_1 : \ldots : \oplus v_N : \textit{eff}$, and the encoding of the object removal effect is equivalent to $\ominus v_1 \wedge \ldots \wedge \ominus v_N$. [6]

**Example 5.** *In our running example, the action buy is written in PDDL as*

```
(: action buy
 : parameters (?L)
 : precondition (headquarters ?L)
 : effect (: new (?T) (at ?T ?L))).
```

This extension simplifies many PDDL models. With standard PDDL, domain experts need to puzzle out how to simulate object creation. This usually involves adding extra predicates, and modifying conditions to take these predicates into account. For example, in the original Settlers domain (Long and Fox 2003) vehicles can be created during the search. To encode this in PDDL, the authors had to introduce a new predicate `potential` that indicated whether some object was a potential vehicle or note. Besides enumerating in the initial state all potential vehicles, they also had to add to every action using an vehicle ?V the precondition (`potential` ?V). The resulting domains are usually much more entangled and complicated than their versions with native PDDL object creation.

## Undecidability Results

We show next that OBJCREATION-PLANEX is undecidable. In fact, it is undecidable even if it contains one single action schema with an object creation effect. We can use our formalism above to decide whether a Turing Machine (TM) halts at a given input or not.

**Definition 3** (Turing Machine). *A Turing Machine (TM) is given by a tuple $M = \langle Q, \Sigma, \delta, q_0, q_{accept}, q_{reject} \rangle$, where*

- *$Q$ is a finite set of states;*
- *$\Sigma = \{0, 1, \square\}$ is the alphabet;*
- *$\delta : (Q \setminus \{q_{accept}, q_{reject}\}) \times \Sigma \to Q \times \Sigma \times \{L, R\}$ is the transition function;*
- *$q_0 \in Q$ is the start state;*
- *$q_{accept} \in Q$ is the accept state;*
- *$q_{reject} \in Q$ is the reject state, where $q_{accept} \neq q_{reject}$;*

*The machine has a head that can move left (L) or right (R), and a working tape. This tape is denoted as Tape. We assume it is infinite to the right, but not to the left. Given an input $x \in \{0, 1\}^*$, Tape starts with $x$ written on its left-most cells, and $\square$ is on all the other (infinite many) cells (denoting that they are empty). The head of the machine starts at the left-most cell (i.e., the first symbol of $x$).*

**Theorem 1.** OBJCREATION-PLANEX *is undecidable.*

---

[6]In PDDL, parameters can be typed. We also allow for this in our extension, but we do not define in our formalism for simplicity.

*Proof.* We reduce the problem of deciding whether a given TM $M$ halts (i.e., reaches either $q_{accept}$ or $q_{reject}$) on a given input $x$ to the problem of deciding if there is a plan for $\Pi$.

The first-order language of $\Pi$ is defined as follows. The variables are $Q_i, S_i, D, C_i$ (also without indices) where $Q_i$ encode states (i.e., $Q_i \in Q$), $S_i$ encodes TM tape symbols (i.e., $S_i \in \Sigma$), $D$ encodes TM directions (i.e., $D \in \{L, R\}$), and $C_i$ encodes TM positions on the tape. It contains the following predicates: *state*$(Q_1)$, encoding that the TM is currently at state $Q_1$; *transition*$(Q_1, Q_2, S_1, S_2, D)$ encoding that from state $Q_1$ when reading $S_1$ there is a transition that writes $S_2$, changes state to $Q_2$, and moves the head in direction $D$; *head*$(C)$ indicating that the head is at cell $C$; *next*$(C_1, C_2)$ indicating that $C_1$ is immediately to the left of $C_2$ in *Tape*; *right-limit*$(C)$, indicating that cell $C$ is the current right-most cell; *symbol*$(C, S)$ encoding that cell $C$ has symbol $S$ written in it; *terminated*, a nullary predicate indicating that the run of the TM terminated in $q_{accept}$ or $q_{reject}$; *is-right*$(D)$ and *is-left*$(D)$ checking if direction $D$ is right (R) or left (L); *accept*$(Q_1)$ and *reject*$(Q_1)$ checking if state $Q_1$ is $q_{accept}$ or $q_{reject}$.

Our task has the following action schemas: (i) read a symbol at cell $C$ and move the head to the left; (ii) read a symbol at cell $C$ and move the head to the right, to a cell that has been reached before or is in the input; (iii) read a symbol at cell $C$ and move the head to the right, to a fresh cell, while expanding the tape; (iv) reject the input; (v) accept the input.

To keep things short, we only show action schema (iii). We denote this action by $A$. It uses variables $Q_1, Q_2, S_1, S_2, C_1, C_2, D$. We define *pre*$(A)$ as

$$pre(A) := state(Q_1) \wedge transition(Q_1, Q_2, S_1, S_2, D)$$
$$\wedge symbol(C_1, S_1) \wedge head(C_1) \wedge right\text{-}limit(C_1)$$
$$\wedge is\text{-}right(D),$$

and the effect *eff*$(A)$ as follows:

$$eff(A) :=$$
$$state(Q_2) \wedge symbol(C_1, S_2) \wedge \neg state(Q_1) \wedge$$
$$\neg symbol(C_1, S_1) \wedge \neg head(C_1) \wedge \neg right\text{-}limit(C_1) \wedge$$
$$(\oplus C_2 : head(C_2) \wedge right\text{-}limit(C_2) \wedge next(C_1, C_2))$$

Action schemas (i) and (ii) are similar, but they have no object creation effect: instead of moving the head to a new tape cell, they use the predicate *next* to encode already existing cells. Action schema (iv) has variable $Q$, precondition *state*$(Q) \wedge reject(Q)$, and effect *terminated*. Action (v) is analogous to (iv) but uses *accept*$(Q)$ in the precondition.

In the initial state $s_0 = \{\mathcal{U}^{s_0}, \{\mathcal{P}^{s_0}\}\}$, the interpretation $\{\mathcal{P}^{s_0}\}$ contains[7]

- *transition*$(q_1, q_2, s_1, s_2, d)$ if $\delta(q_1, s_1) = (q_2, s_2, d)$;
- *state*$(q_0)$;
- for the $i$-th symbol of the input $x$, we introduce the atom *symbol*$(c_i, s_i)$ where $c_i$ is the $i$-th cell of *Tape* and $s_i \in \{0, 1\}$ is the respective symbol in $x$. If a cell $c'$ comes after (i.e., to the right) of $c$ in *Tape*, we introduce predicate *next*$(c, c')$. If $c$ is the last cell in *Tape*, we add *right-limit*$(c)$.

---

[7]Again, using atoms to write $\{\mathcal{P}^{s_0}\}$ for simplicity.

**Procedure 1**: Compute plan for tasks with object creation

1: $\mathcal{S} \leftarrow \emptyset$
2: openList $\leftarrow \{s_0\}$
3: **while** openList $\neq \emptyset$ **do**
4:     $s \leftarrow$ openList.Extract()
5:     **if** $s \models G$ **then return** plan
6:     **for all** $A \in \mathcal{A}$ **do**
7:         **for all** $\sigma : \textit{free}(A) \to \mathcal{U}^s$ **do**
8:             **if** $s, \sigma \models \textit{pre}(A)$ **then**
9:                 openList $\leftarrow$ openList $\cup \{\text{succ}(s, \sigma, A)\}$
10: **return unsolvable**

- $\textit{head}(c_0)$ where $c_0$ is the first cell of the tape,
- $\textit{is-right}(\texttt{R})$ and $\textit{is-left}(\texttt{L})$,
- $\textit{accept}(q_{accept})$ and $\textit{reject}(q_{reject})$,

while $\mathcal{U}^{s_0}$ contains all objects in $\{\mathcal{P}^{s_0}\}$. The goal is defined as $G = \textit{terminated}$.

The initial state of our task exactly encodes the initial configuration of $M$ in input $x$: the head starts at the first cell, the *state* is the initial state $q_0$, and the input is encoded in the first cells of the tape. The actions simulate the possible transitions between configurations of the TM. The important detail is that we initialize only a finite number of *symbol* predicates, which is actually the length of $x$. Whenever we need to use new cells, we can use the action schema described above to append an extra cell at the end of the tape.

Task $\Pi$ can simulate $M$ precisely. If there exists a plan for $\Pi$, the plan can be converted into a trace of configurations of $M$. Similarly, if $M$ never halts, $\Pi$ has no plan. $\qquad\square$

Theorem 1 shows that planning with object creation is undecidable in general. However, when plans exist we can still compute them. In other words, OBJCREATION-PLANEX is semi-decidable.

First, consider Procedure 1. It shows a general state-space search (without duplicate elimination). It works just the same for tasks with object creation. If openList behaves as a FIFO, then Procedure 1 is a breadth-first search. This procedure can also be extended to accommodate heuristic estimates or other sort of optimizations.

**Corollary 2.** OBJCREATION-PLANEX *is semi-decidable.*

For Corollary 2, note that although the state space is infinite, we are searching for a finitely long path in a finitely branching state space. For such scenarios, breadth-first search (i.e., Procedure 1 using a FIFO open-list) is semi-complete because it will consider all (finitely many) paths of length $k$ before considering any longer path. Hence, if a solution exists, it will be found after a finite computation

**Corollary 3.** OBJCREATION-PLANLEN *is decidable.*

To see this, we can again run breadth-first search, keeping track of the length of generated paths and rejecting the input as soon as we reach the given length bound.

## Overall Procedure in Practice

But there are still some details missing for a practical implementation of the above procedure. For example, we would like to quickly generate successor states, and also to define how to come up with fresh objects during object creation.

To generate successor states, we need to find all assignment functions leading to applicable actions. Let $s$ be a state and $A$ an action schema. We can find all ground actions $A^{s,\sigma_1}, \ldots, A^{s,\sigma_m}$ by computing all $\sigma_i$ such that $s, \sigma_i \models \textit{pre}(A)$. As states are finite, solving this problem is decidable. Free variables occurring in $\textit{eff}(A)$ but not in $\textit{pre}(A)$ can take any possible value, so we consider all possible assignments.

Now assume that $A$ has an object creation effect $\oplus v : e$, where $e$ is an effect. At a given state $s$, we need to instantiate $v$ to a fresh object that is not in $\mathcal{U}^s$. There are infinitely many ways to do so. The new object could be assigned to a natural number, or it could be an arbitrarily long sequence of characters, like $\texttt{aaaa}$, or anything else that is not in $\mathcal{U}^s$. But all these choices are just names assigned to the new object, and they do not influence the semantics of the successor state. In other words, they are just syntactic. Any such choice of name is *isomorphic* to the other ones. Choosing one well-defined method to come up with names is sufficient. We call a function that chooses the next fresh object in a given state a *choice function*.

One of the simplest ways is to map every object $o \in \mathcal{U}^{s_0}$ to an index $\text{id}(o) = i$ for $i \in \mathbb{N}$. Whenever we need a fresh object in a state $s$, we compute the minimum $j \in \mathbb{N}$ not assigned to any object in $\mathcal{U}^s$. We then introduce a new object named $j$ and set $\text{id}(j) = j$. In other words, new objects are identified by the minimum non-used index in the current state.[8] Removing an object $o$ unassigns $\text{id}(o)$. A successor state keeps the same mapping as its parent state, besides the newly created or removed objects. For example, if our state $s$ has three objects, we can map them to indices 1, 2, and 3. If the action $A^{s,\sigma}$ has an object-creation effect, this object can be assigned to index 4. The successor state $succ(s, \sigma, A)$ still maps the three original objects to 1, 2, and 3, but it also maps the fourth object to 4.

This bring us to yet another efficiency concern. Assume that we have a state $s$ and two actions $A$ and $B$. Let us also assume that $A$ and $B$ have an always satisfiable precondition. Moreover, $\textit{eff}(A) := (\oplus v : P(v))$, while $\textit{eff}(B) := (\oplus v : Q(v))$. The sequences $\langle A^{s,\sigma_1}, B^{succ(s,\sigma_1,A),\sigma_2} \rangle$ and $\langle B^{s,\sigma_2}, A^{succ(s,\sigma_2,B),\sigma_1} \rangle$ lead to two different states, but both are semantically equivalent: they only differ by the names used to identify the created objects. The two resulting states are *isomorphic*, and keeping only one of them is sufficient.

State-space search algorithms usually rely on duplicate state detection, but this is not enough here because we want to detect all *isomorphic* states. Unfortunately, no polynomial-time algorithms are known for this (Grohe and Schweitzer 2020).

---

[8] This requires keeping track of the id-value for created objects as we process the effect of an action. For a choice function where this is not necessary, see the supplementary material.

This problem is similar to the one faced by orbit space search algorithms (Alkhazraji et al. 2014; Domshlak, Katz, and Shleyfman 2015). In orbit space, search nodes correspond to equivalence classes of states, instead of individual states. Two states are considered equivalent if they are detected to be symmetric. This symmetry detection is usually done based on *canonical states*.

Ideally, a canonical state would be a unique representative of an equivalence class. During search, it is sufficient to store the canonical state for each encountered equivalence class and then use standard duplicate elimination techniques. The efficiency of canonical state computation is a crucial part of the performance of orbit space search planners. In practical implementations, computing true canonical representatives is considered to be too expensive, and therefore canonical states are approximated by a greedy procedure. This leads to some lost opportunities for detecting equivalence, but does not affect correctness.

In planning with object creation, one would expect symmetrical states to occur often, as the different names given to new objects are another source of symmetry. We can tackle this problem as in orbit search, by using (exact or approximate) canonical states for each equivalence class.

**Example 6.** *Let us say we have two states $s_1$ and $s_2$, and $a, b, c$ and $d$ are objects created during search:*

$$\mathcal{U}^{s_1} = \{a, b, c\}, \quad \mathcal{P}^{s_1} = \{P(a), P(c), Q(a), Q(b)\},$$
$$\mathcal{U}^{s_2} = \{b, c, d\}, \quad \mathcal{P}^{s_2} = \{P(b), P(d), Q(b), Q(c)\}.$$

*These states are equivalent via the object mapping $\{a \mapsto b, b \mapsto c, c \mapsto d\}$. In this case, this would be already detected by a very simple algorithm approximating canonical representatives by mapping each object to its index in a lexicographical order: for $s_1$ we would map $\{a \mapsto 1, b \mapsto 2, c \mapsto 3\}$, and in $s_2$ we would map $\{b \mapsto 1, c \mapsto 2, d \mapsto 3\}$. In both cases, we would end up with the same state $s'$, showing equivalence:*

$$\mathcal{U}^{s'} = \{1, 2, 3\}, \quad \mathcal{P}^{s'} = \{P(1), P(3), Q(1), Q(2)\}.$$

## Decidability for State-Bounded Tasks

Inspired by the work on bounded situation calculus (De Giacomo, Lespérance, and Patrizi 2016), we consider the case in which the number of objects in $\mathcal{U}^s$, for any state $s$, is bounded by a fixed constant $k$. We say that such a task is *state-bounded*, with bounding constant $k$.

There are several sufficient conditions to guarantee state-boundedness (De Giacomo, Lespérance, and Patrizi 2016). We refer the reader to the original paper for details.

**Theorem 4.** *If the domain is state-bounded, then* OBJCREATION-PLANEX *is decidable.*

This case is decidable by direct reduction to situation calculus (e.g., Claßen, Hu, and Lakemeyer 2007) using the results in De Giacomo, Lespérance, and Patrizi (2016). Let us first define a *recycling choice function*: a choice function is said recycling if, after an object is removed from a state $s$, the object can be re-introduced by an object creation effect in some state reachable from $s$. The choice function described

above assigning every object to a natural number is recycling. Intuitively, if we use a recycling choice function we create only finitely many new objects. Hence, Procedure 1 considers only finitely many states, and it is thus decidable. But note that the state-space of the task is still infinite, although each state has bounded size.

## Implementation

We extended the Powerlifted planner (Corrêa et al. 2020) to allow object creation effects. The source code of our implementation and our benchmarks will be publicly available.

Powerlifted is a heuristic search planner that only supports STRIPS (Fikes and Nilsson 1971) extended with types, so we only implemented a STRIPS fragment of our formalism. We consider the following as a *STRIPS task with object creation*: preconditions are restricted to conjunctive formulas over positive atoms; an effect can only be a simple, conjunctive, or object-creation effect; the goal is a conjunctive formula. Note that Theorem 1 uses this fragment in the proof, so plan existence for STRIPS tasks with object creation has the same decidability results as the complete formalism.

In this fragment, the algorithm by Corrêa et al. (2020) is still sufficient to compute the applicable ground actions. As preconditions are conjunctions of positive atoms, the precondition of an action $A$ can be considered as a conjunctive query (Ullman 1989). If we answer this query over a state $s$, every tuple in the answer corresponds to a function $\sigma$ mapping $free(A)$ to $\mathcal{U}^s$ such that $A^{s,\sigma}$ is applicable. Corrêa et al. (2020) exploit structural properties of these queries (i.e., acyclicity) to compute successor states efficiently.

We use the choice function mapping objects to natural numbers as described above (i.e., a new object $o$ is mapped to the smallest natural number $j$ such that there is no $o'$ where $\text{id}(o') = j$). We did not implement any isomorphism check between states, and we rely on syntactic duplicate detection. We leave more sophisticated techniques based on (approximate) canonical representatives as future work.

To improve the search, we modified the lifted width-based search (Lipovetzky and Geffner 2012, 2017) implemented by Corrêa and Seipp (2022). Best-first width search (BFWS) (Lipovetzky and Geffner 2017) uses the *novelty* measure to choose which states to expand. The novelty $w(s)$ of a state $s$ is the size of the smallest non-empty set of ground atoms $Q$ such that $s$ is the first state visited where $s \models Q$. For example, if $s$ is the first state containing atom $a$, then $w(s) = 1$. In contrast, if there is no single atom that first occurred in $s$ but there is a subset $\{a, b\}$ that first occurred together in $s$, then $w(s) = 2$. A more informed version of novelty is $w_f(s)$, which is computed only considering tuples in states $s'$ where $f(s) = f(s')$. We implemented BFWS with $w_{\#G}(s)$, where $\#G$ is the number of atoms in the goal satisfied in $s$. We only compute $w_{\#G}$ up to pairs. If there is no new pair in $s$, then $w_{\#G}(s) = 3$.

Novelty measures do not seem to fit with object creation: introducing a fresh object makes the state have a novelty of 1, so BFWS always prioritizes states that create objects. In domains where the number of created of objects is unbounded, this could lead to an infinite sequence of actions. To solve this, our implementation only consider tuples of

atoms that do not mention new objects. In other words, we compute the novelty of a state over those tuples that only mention objects in $\mathcal{U}^{s_0}$. However, this has the completely opposite effect: new objects do not account for the novelty of a state, so they do not add any information to the BFWS. What can happen in this case is that BFWS finds plans creating the minimum number of objects necessary. As we will see in our experimental results next, this does indeed happen often. Yet, this modified BFWS still improves our planner.

## Experimental Results

Our experiments were run on an Intel Xeon Silver 4114 processor running at 2.2 GHz using a runtime limit of 30 minutes and a memory limit of 8 GiB per task.

We use four PDDL domains with object creation in our benchmarks. Two of them are based on previously existing domains that encode object creation by listing all possible objects at the initial state and using auxiliary predicates to simulate object creation. This made it necessary in the original PDDL to introduce an arbitrary on the number of objects that can be created and then experiment with different bounds to deal with tasks that are wrongly considered unsolvable because the number of objects is too low.

For each domain, we have two versions: one using our new PDDL extension, and one where all potential objects are declared in the initial state. The version using the new syntax is called the *extended version*, while the other is called *standard version*. This comparison is imperfect because only the extended version captures the underlying problem faithfully, but it allows us to compare our planner using object creation with existing planning systems.

**Cluster Management**   In this new domain, we must produce a set of files. These files are produced by executing scripts on certain inputs. For example, executing script $S$ with input $I_1$ might output $O_1$, and executing $S$ with $I_2$ might output $O_2$. Our benchmark contains instances with up to 100 files and 20 different scripts. We also have a cluster with multiple CPUs, where we can load and execute these scripts with the corresponding files. The actions are to load a file or script in a CPU, to execute a script, to save a file into memory, and to add a new CPU to the cluster. So if a script must be used several times (with different inputs to produce different outputs), it might be preferable to load it only once in one of the CPUs and leave it there. The problem in this domain is to find the optimal amount of CPUs to obtain a certain set of goal files as quick as possible. We can add new CPUs to our cluster using an action that creates a new CPU object. In the standard version, we pre-declare 5 CPUs.

**Commutative Rings**   This domain was introduced by Petrov and Muise (2023). A task in this domain is a statement in elementary algebra. A plan is a proof for this statement. The domain focuses on tasks related to commutative rings. One task, for example, is to prove that for all commutative rings $R$, $a \times 0 = 0$ for every $a \in R$. Action schemas represent the axioms of commutative rings, equality operations, definitions of products, sums, and inverses. Object creation can be used to model existential axioms and build complex expressions. For instance, given a commutative ring $R$,

for any $a, b \in R$ there exists an element $a+b \in R$. So we can construct a new object $c$ and define $c = a+b$ to be used later. In the original domain, Petrov and Muise (2023) introduce a fixed number of undeclared variables at the initial state. However, this makes the task harder to ground, while also bounding the number of proofs the planner can explore. We used the original tasks of this domain, but compiled away conditional effects, which are not supported by Powerlifted.

**Logistics Company**   This is a new domain, and it is similar to the running example in our paper. We have a set of connected locations and a set of packages that must be delivered to specific locations. Our company has headquarters in a few locations, and we can buy trucks that appear at one of these headquarters. Actions are to move the truck, (un)load packages, and buy a new truck. The challenge is to find a good balance between how many trucks to buy to deliver all packages efficiently. While all tasks are solvable with one single truck, it might be that using multiple trucks decreases the plan length significantly. In our instances, the number of locations varies between 3 and 1 000, the number of packages from 1 to 100, and the number of headquarters from 1 to 20. In the standard version, the number of declared trucks at the initial state is twice the number of headquarters in the task.

**Settlers**   This domain is based on the Settlers domain used in the IPC 2002. (Long and Fox 2003). The domain focuses on resource management. Products and factories must be built from raw materials and used in the manufacture or transportation of further materials. The objective is to construct a variety of building types at various specified locations. The original domain is numeric: the quantity of resources at each location is defined by numeric fluents. But these fluents are discrete and their maximum values are always bounded, so one can emulate them using predicates to encode a successor relation over the natural numbers. Long and Fox (2003) mention that this domain highlights the necessity of object creation during plan execution. In the standard version, all objects had to be declared in advance, which made grounding harder and made the modeling more convoluted. We removed "maritime" objects – wharfs, docks, ships – in our version, because the original instances were too challenging for all planners.

We ran Powerlifted on both versions of our benchmarks (standard and extended versions). Powerlifted using the standard versions is denoted as PWL, and using the extended versions as PWL$^{++}$. For each, we tested two configurations: a breadth-first search (BFS), and the BFWS as explained in the previous section. These are not the best configurations of Powerlifted (e.g., Corrêa et al. 2022), but are the best-performing configurations that we could extend to support object creation within the scope of this paper.[9]

Table 1 shows the coverage of all methods in our benchmark. PWL$^{++}$ outperforms PWL when using BFWS, but it has lower coverage with BFS. The BFWS implementation does not consider tuples considering newly created objects.

---

[9]It would be interesting to extend delete-relaxation heuristics as implemented in Powerlifted to object creation, but this is not a simple task and requires developing its own theory.

| | PWL++ | | PWL | | FD | |
|---|---|---|---|---|---|---|
| | B | W | B | W | B | W |
| Cluster Man. (20) | 3 | 10 | 2 | **14** | 5 | 12 |
| Comm. Ring (15) | 2 | 10 | 9 | **14** | 8 | 10 |
| Logistics Comp. (20) | 3 | **18** | 5 | 8 | 5 | 6 |
| Settlers (20) | 3 | **8** | 3 | 6 | 3 | 4 |
| **Total** (75) | 11 | **46** | 19 | 42 | 21 | 32 |

Table 1: Coverage of PWL++, PWL, and FD on our benchmark. For each planner, we tested a configuration with breadth-first search (B) and best-first width search (W).

This can be an issue with the extended versions, as only some objects are considered when evaluating the novelty of a state. But this is not always problematic. In the Logistics Company domain, for example, using one truck is already enough to solve the task so BFWS does not favor the creation of new trucks. This increases coverage in the extended version but also impacts plan quality. In some cases, the optimal plan had length 10, but PWL++ with BFWS only found plans with more than 100 steps.

With respect to runtime, PWL++ and PWL are comparable in both configurations. With respect to memory, there are larger differences depending on the domain. In a few instances of the Commutative Ring, PWL++ used up to 10 times more more memory than PWL. In this domain, some of the original instances in the standard version do not use any undeclared variables. In the extended version, the planner is not aware that they are not needed, so they are introduced multiple times. This blows up the size of the state space. To solve this problem it is crucial to have heuristic estimates that can better decide when more objects are needed.

Powerlifted has some advantages when solving tasks that are hard to ground, but its search capabilities are not on par with ground planners. To compare PWL++ with a state-of-the-art heuristic search planner, we also ran Fast Downward (Helmert 2006) on the standard versions. To keep the results comparable, we used the blind search and we implemented BFWS in Fast Downward. We denote Fast Downward as FD in the rest. Results are also shown in Table 1.

When using blind search, FD is significantly better than PWL++ and somewhat better than PWL. This is consistent with previous results (Corrêa et al. 2020). For larger problems, grounding becomes a challenge (Corrêa et al. 2023). This can be seen when comparing FD with BFWS and PWL++ with BFWS. Although the coverage of FD increases by almost 50% when switching from blind search to BFWS, it is still worse than PWL++. Only in the Cluster Management domain, FD outperforms PWL++. This is also the only domain where all tasks can be grounded within seconds, as the number of ground actions is low (a few thousands) even declaring all objects in advance. In the Logistics Company domain, where several of the declared objects are not necessary (although helpful) grounding does become a major bottleneck. In this domain, FD has worse coverage than PWL++. The same happens in the Commutative Ring domain, even though in this domain we have at most one undeclared variable per task – but in the grounding several actions use this object. As noted by Petrov and Muise (2023), adding one single undeclared variable to the initial state is already enough to make the grounding much harder.

## Related Work

Object creation has been considered several times as an important feature to large scale planning systems (Long and Fox 2003; Petrov and Muise 2023). So far, all methods trying to solve this problem used compilations. Fuentetaja and de la Rosa (2016) present an automatic compilation of "irrelevant objects" (i.e., objects whose name do not specifically matter) into counters. While this is all right in certain domains, it has inherent limitations. Two irrelevant objects can only compiled into the same counter if they are fungible and do not differ their properties during the plan. In our logistics company example this is impossible, as trucks do differ (e.g., each truck can be at a different location). Counters also must be first compiled into relations, which forces them to be bounded. Similar compilations have been proposed to automatically encode indistinguishable objects into counters represented by numeric variables (Riddle et al. 2016).

In situation calculus (McCarthy 1963), infinitely many objects have already been considered (cf. Reiter 2001) although mostly in theory. In comparison to our work, the most relevant result in situation calculus is the work by De Giacomo, Lespérance, and Patrizi (2016). They show that bounded situation calculus – when the number of objects in tuples is bounded – is decidable. As aforementioned, this is essentially the same as planning with object creation for tasks with bounded number of objects.

Lifted planners were dominant in the 1990s (Penberthy and Weld 1992; Younes and Simmons 2003). Although progress in this area was slow during the early 2000s, recent work showed its usefulness in hard-to-ground benchmarks (Corrêa et al. 2021; Lauer et al. 2021; Wichlacz, Höller, and Hoffmann 2022; Höller and Behnke 2022; Shaik and van de Pol 2022; Horčík, Fišer, and Torralba 2022). We used the Powerlifted planner, but other lifted heuristic search planners should be capable of dealing with object creation (e.g., Horčík and Fišer 2021).

## Conclusion

We formalized an extension of classical planning that allows for object creation and removal during plan execution, called "planning with object creation" for short. In our formalism, this creation/removal happens as special kind of action effects. In general, planning with object creation is semi-decidable; when states are bound, however, the problem becomes decidable. We also implemented support for object creation/removal in the Powerlifted planner (Corrêa et al. 2020). In our experimental results, support for this extension caused no harm to the planner performance. It was also on par with state-of-the-art classical planners (Helmert 2006). To further scale performance, one could use more informed heuristic estimates for tasks with object creation.

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
