# OpenReview forum: "Planning with Object Creation"
_icaps-conference.org/ICAPS/2024/Conference — ICAPS 2024_

### Official Review · Reviewer_wJuQ · 2024-01-15

**Significance And Importance:** 2
**Soundness:** 2
**Novelty:** 2
**Clarity:** 3
**Confidence:** 3

**Weaknesses:**

0: Minor weaknesses requiring some work to be addressed for the paper to be accepted.

**Contributions Of The Paper:**

The paper introduces an extension to classical planning frameworks by allowing action effects to create and remove objects during planning, addressing the limitations of fixed object sets in traditional planning formalisms, such as PDDL.

The key contributions of the paper can be summarized as follows:

1.	Extension of Planning Formalism: Introducing an extension to classical planning that permits dynamic creation and removal of objects during the planning process, which significantly simplifies domain encoding and reduces the dependency on expert knowledge.

2.	Theoretical Insights: Providing theoretical insights into the computational aspects of planning with object creation. Establishing that while classical planning with object creation is semi-decidable, ensuring plan existence, the problem becomes decidable for optimal planning and bounded state tasks, even with an infinite state-space.

3.	Practical Implementation and Tool Extension: Developing a PDDL extension to accommodate object creation and removal in action effects. Extending the Powerlifted planner, a state-of-the-art tool, to handle this new formalism, showcasing improved performance compared to traditional encodings.

4.	Performance Enhancement and Utility: Demonstrating through practical implementation that allowing object creation and removal during planning enhances planner efficiency by reducing unnecessary objects and state size, making planning more effective across various domains.

**Ethical Considerations:**

(1) Not Applicable: The paper does not have any ethical considerations to address

**Nomination For Best Paper:**

No

**Overall Evaluation:**

-1: (weak reject)

**Questions For Authors:**

Q1 : Why differentiate between objects and constants? Both concepts stem from PDDL, yet from a theoretical and logical standpoint, they essentially represent constants. Focusing solely on constants might offer greater clarity in discussions.
Q2: In the evaluation, two versions of each domain were used: one based on the introduced formalism allowing object creation and removal, and the other where all potential objects were declared upfront. The selection process for potential objects in the latter version might not have been explicitly detailed in the paper.
Typically, the set of potential objects in the second version (where all potential objects are declared) involves identifying the minimal set of objects needed to find a solution. This set could include objects required by the actions, initial state, and goal specifications within the domain.
In some cases, additional predicates might be introduced, similar to the original Settlers benchmark proposed by Long, to simulate or encompass potential objects that might be dynamically created or removed in the formalism allowing object manipulation.
However, the paper might benefit from providing a more explicit and detailed explanation of how the second version of the domains, with all potential objects declared, was produced. This could involve specifying the criteria used for object selection, potential predicate additions, and any assumptions made to simulate the object set.
Adding such details would offer transparency and clarity, ensuring a better understanding of the process used to create the comparison versions of the domains.
Could you detail this point?
Q3: Why just consider coverage criteria in evaluation and not plan quality ?
Q4: Why not compared you approach with compilation ones ? Is there any reason why this comparison cannot be made?

**Reproducibility:**

4: Authors promise to release code and domains (whichever apply).

**Strengths Of The Paper:**

1.	The paper offers diverse and compelling contributions to the community.
2.	The paper is effectively written, presenting ideas in a pedagogical manner that facilitates easy comprehension.
3.	The paper's theoretical findings regarding decidability and existence are highly pertinent and engaging for the community.

**Weaknesses Of The Paper:**

1.	The paper's contributions could benefit from a clearer structure.
2.	The experimental evaluation of the approach has room for further development and expansion.
3.	The experimental evaluation of the approach might benefit from streamlining domain presentations while focusing more on illustrating the quality of solution plans or compared the proposed approach with existing compilation approaches. Additionally, expanding the test set to include more domains could enrich the evaluation.
4.	While the related work section touches upon plan generation with object creation, it appears relatively brief and somewhat incomplete in capturing all directly relevant literature in this domain. It would have been interesting to have more details on compilation approaches such as “Introducing Dynamic Object Creation to PDDL Planning ICAPS 2019 Workshop WIPC.

---

> ### Author Rebuttal · Authors · 2024-01-26
>
> Thanks for the comments and questions!
>
> 1) In logic, constants are syntactic entities and objects are semantic (part of
> interpretations). In standard PDDL, they are in 1:1 correspondence, but this
> changes with object creation/removal. E.g., if constant "c" is part of an action
> precondition and object removal deletes it, we can't delete the logical constant
> because then the action precondition no longer is a well-formed formula. There
> are other ways to deal with this (e.g., standard names); ours is the cleanest we
> found after some iterations.
>
> 2) Fair point. For 2 of the 4 domains, the "potential object" formulations
> already existed, so we used them as-is. For the others (logistics
> company/cluster management), we tried to use a "reasonable" number of objects in
> the spirit of the other domains. It would indeed be nice to give results for
> varying numbers of objects. As stated in the paper, we will publish the
> benchmarks, which should help with transparency. We can also extend the
> discussion of the design decisions if desired.
>
> 3) Because of space, but also there aren't many interesting trends to see.
> We mention the main takeaways on plan quality in the text, but can elaborate.
> (Please also see answer to Reviewer ph6v.)
>
> 4) Regarding compilation, we assume you mean A) the Fuentetaja & de la Rosa
> paper we mention and B) the "Introducing Dynamic Object Creation to PDDL
> Planning" WIPC submission you mentioned.
>
> For A), we discuss in "Related Work" that their approach is limited to fungible
> objects. To elaborate: the compilation needs scenarios where it is sufficient to
> count how many objects are in each "substate", where a substate can only depend
> on a fixed combination of object parameters. For example, vehicles in the
> Logistics Company domain do not have this property because two vehicles can
> carry arbitrarily different sets of packages, so we would need a function taking
> a set of packages as an input (impossible in PDDL). The same holds for all
> domains we consider. Therefore, the Fuentetaja & de la Rosa approach doesn't
> apply to these domains.
>
> For B), we are aware of the submission. However, it was rejected from the
> workshop (which we confirmed with the workshop chairs) and is only public
> because of the way OpenReview was set up for WIPC. We were advised it would be
> inappropriate to discuss a paper with such a status because it could hurt the
> authors' chances to have a revised version of the work published. In any case,
> the approach is different from ours.

---

### Official Review · Reviewer_KB2G · 2024-01-17

**Significance And Importance:** 3
**Soundness:** 4
**Novelty:** 3
**Clarity:** 3
**Overall Evaluation:** 2
**Confidence:** 4

**Weaknesses:**

2: No major or minor weaknesses.

**Contributions Of The Paper:**

The paper introduces a new technique for planning with object creations, it proves that planning with object creation is semi-decidable and provides a clear and sufficient experimental analysis. The paper is a starting point for future work, and I believe it paves the way for the idea of "Planning with Object Creation" to be better improved in the future.

**Ethical Considerations:**

(1) Not Applicable: The paper does not have any ethical considerations to address

**Nomination For Best Paper:**

No

**Questions For Authors:**

1) For me, it was confusing Theorem 1 stating that "OBJCREATION-PLANEX is undecidable" and Corollary 2 stating "OBJCREATION-PLANEX is semi-decidable". How can they be both?
2) At line 360 you state that "we compute the minimum j ∈ N not assigned to any object in Us". This computation is pointless, right? It can be simply stored somewhere and retrieved. Computing the minimum makes it look like a hard task, but for me, it should be a simple look-up table.
3) Why do you only deal with object-creation in the experimental analysis and not with object-removal?
4) In the cluster management domain, at line 535 you state the need to find "the optimal amount of CPUs". The word "optimal" here is not the right choice, since you do not make optimal planning, right?

**Reproducibility:**

4: Authors promise to release code and domains (whichever apply).

**Strengths Of The Paper:**

The paper is well written. The humorous introduction made me smile, perhaps it is a little bit informal, but it sold the main point effectively.
In general, the main idea of the paper is good, and I believe it could really be helpful in practical scenarios. I really appreciated the notational and background work, together with the proofs of semi-decidability which, I believe, will pave the way to future work. The experimental analysis is sufficient (more on that on the weaknesses section).

**Weaknesses Of The Paper:**

- The paper itself, on a practical point of view, doesn't introduce anything regarding how to better solve planning with objects creation. The   "Overall Procedure in Practice" and "Implementation" section reuse some existing techniques and solvers to make it work with object creations. For me, this is not a big weakness, as stated in the strengths section, I believe this paper aims to introduce the paper from a theoretic point of view and I believe more work will come afterwards to push the envelope further. And for me, this is sufficient for accepting the paper.
- I believe that Table 1 should also include the times and the memory, since you claim PWL++ to be more memory expensive.
- At line 631 you call B as "blind" while it is "breadth-first search" (although in A* they mean the same thing)

Typos:
- Third bullet point after line 170, it should be i \in \{1,2\}
- Line 228: "note" -> "not"
- Line 230: "an vehicle" -> "a vehicle"
- Line 590 should be another paragraph

---

> ### Author Rebuttal · Authors · 2024-01-26
>
> Thanks for the comments and useful suggestions!
>
> 1) Undecidable means "not decidable", and "semi-decidable" is a weaker condition
> than "decidable". The halting problem is undecidable, but also
> semi-decidable. In general, a problem is decidable iff it is semi-decidable
> and its complement is semi-decidable.  In practical terms, it is possible to
> write an algorithm that solves all solvable planning tasks with object
> creation (semi-decidable), but it's impossible to write an algorithm that
> also reports unsolvability on all unsolvable inputs. (Instead, it would run
> forever on some unsolvable inputs.)
>
> 2) Yes, storing the value in each state is sufficient and what the implementation actually
> does. (A recomputation can become necessary after an object deletion.)
>
> 3) We went for domains that were available in the literature or created
> domains with similar features to what others have requested (see referenced
> papers). It seems that from what we've seen so far, there is a larger demand for
> creation than deletion.  The reason is perhaps that, unlike creation, deletion
> can easily be simulated -- just mark an object that is not to be used any more
> with a special predicate, or conversely mark the "active" objects that may be
> used. Indeed in preliminary discussions we had with other members of the
> community, the discussion was always on object creation; we don't recall someone
> saying "wouldn't it be useful if you could delete objects in PDDL?".  We
> primarily include deletion because it seems asymmetric not to do this.
>
> 4) Right, this was poorly phrased. We will explain what the objective function
> determining plan quality is, but a plan does not have to be optimal
> w.r.t. number of CPUs to be a solution.

---

### Official Review · Reviewer_ph6v · 2024-01-23

**Significance And Importance:** 2
**Soundness:** 3
**Novelty:** 2
**Clarity:** 3
**Confidence:** 4

**Weaknesses:**

0: Minor weaknesses requiring some work to be addressed for the paper to be accepted.

**Contributions Of The Paper:**

The paper proposes to extend the action effects in planning to create and remove objects in the context of lifted planning. Using lifted planning is relevant here, since the objects to instantiate the actions at every state depend on the actions applied to reach that state, and generating a static instantiation at preprocessing is not enough in this setting.  The extension was implemented in the Powerlifted planning system (for classical planning). The paper describes the planning formalism from the point of view of first-order logic, some theoretical results showing that the plan existence decision problem is undecidable and semi-decidable; and that it is decidable when the plan length or the number of objects is bounded. It also includes some experiments in four domains showing that in two of the domains the coverage is improved when using best-first width search, but the quality of the plan can become much worse.

The contribution is moderate. The extension is rather straight-forward. The theoretical results are not surprising, can they be derived from previous work? (Turing Machine by Bylander 94) and that the decidability of those problems is known. The important problems here (duplicate detection, heuristics) are not addressed in the paper. There is some theory on duplicate detection but this is not evaluated in practice.

**Ethical Considerations:**

(5) Excellent: The paper comprehensively addresses all of the applicable ethical considerations

**Nomination For Best Paper:**

No

**Overall Evaluation:**

-1: (weak reject)

**Questions For Authors:**

1. Can you comment on how the order of effects is considered here?
2. Can you clarify the issue with the successor generation?
3. Can the results be derived from the Turing Machine by Bylander 94?
4. Can you include experiments on quality and expanded states?

**Reproducibility:**

4: Authors promise to release code and domains (whichever apply).

**Strengths Of The Paper:**

The fact that planning actions can not create/remove objects is a very important limitation in planning. Estimating the number of objects required to solve a planning task in advance is a very difficult task and defining a very large number of objects can have a high negative impact in the planners' performance. This is well motivated by the authors.

It is a problem that has yet to be addressed at some point, and it makes sense to address it for lifted planning.

**Weaknesses Of The Paper:**

Main one is that I see the contribution as quite moderate.

It is not specifically stated in the paper but it seems this work is for classical planning with negative and disjunctive preconditions and
forall and conditional effects, without action costs?. This should clearly stated. The implementation is for classical planning (strips planning with types).

The paper says nothing about how the order of different effects with creation/removal affects the action application. But with forall effects the order of effects is now relevant, and this does not seem to be considered by the "changes" function (conjunction of effects is formalized with unions). For instance, the next state for an action with the effects: (1) create truck t with p(t) (2) create a fact q for all trucks would be different if the effects are considered in the reverse order: (1) create a fact q for all trucks, (2) create truck with p(t).

Definition of successor state: formalization of P' is weird and it seems incorrect to me. With the current formalization it seems that P' is a set of objects instead of a set of object tuples. I imagine the authors want to say that the new object tuples in P' are restricted to those that contain objects that have not been remove, which is the same as removing from the previous state all facts concerning a removed objects.

Introducing dynamic creation/removal of objects in PDDL is rather straight-forward. The authors say that they not define types in the formalism for simplicity, but in PDDL is very easy to define a syntax for introducing new objects with types. I don't see any reason not to
include them in the paper.

Undecidablity results:  is there any relation with the following works?:
The computational complexity of propositional STRIPS planning. T. Bylander. Artif. Intell. 1994.
Complexity, decidability and undecidability results for domain-independent planning. K. Erol, D.S. Nau, V.S. Sybrahmanian. Artif. Intell. 1995

line 95: that a formula -> that a formula $\varphi$
line 100: s undefined. Replace it by $\mathcal{I}$ or define s
line 124: The goal is to have p1 at c2 (to move is an action)

-------------------
Post-rebuttal
-------------------
Thank you for your response.
Some previous works on planning for web service composition was mentioned in the discussion. Specifically, a JAIR 2009 paper Hoffmann, Bertoli, Helmert and Pistore contains a very similar undecidability proof.  This should be revised and cited.

---

> ### Author Rebuttal · Authors · 2024-01-26
>
> Thanks for the comments and corrections!
>
> 1. Order of effects does not matter. Our extension follows the conventions of
> existing PDDL: all effects are evaluated with the current state as context.
>
> Consider the effects:
>
> (i) (Q(c) and (forall ?x : if Q(?x) then P(?x))
>
> (ii) (forall ?x : if Q(?x) then P(?x)) and Q(c))
>
> Without knowing PDDL semantics, it might appear (i) always sets P(c) because it
> "first" adds Q(c) and "then" adds P(?x) for all ?x where Q(?x) holds. With such
> a semantics, (ii) would behave differently from (i). But this is not how PDDL
> works: everything is evaluated in the current state, so we only add P(c) if Q(c)
> is true when the action is applied.
>
> The changes function recursively collects all changes and applies all at
> once. In your example, the successor contains a new truck t with p(t) and all
> other trucks t' have q(t').
>
> 2. There was indeed a typo in the definition of P', thanks! The "t" in the
> definition is meant to be a tuple, but then our use of "t in U'" doesn't make
> sense. The fix is to be explicit what is/isn't a tuple:
>
> P' = { <t1, ..., tn> \in (P^s \ Del^P) \cup Add^P | t_1, \ldots, t_n \in U'}
>
> 3. TM reductions are common to many planning variants (e.g. Rintanen ICAPS 2004
> for partial observability). Bylander's results don't apply; they deal with
> propositional planning and only go up to PSPACE. The Erol et al. results,
> building on Chapman 1987, are more closely related but don't entail our results
> because they use a more general language with infinitely many objects in a
> single state. We can say more on this in the paper.
>
> 4. We are happy to use an extra page to discuss it (and will also publish the
> raw results). Summary: width search produces longer plans on average, consistent
> with literature. (The blind baseline algorithm is optimal.)  State expansions
> show the same trends as coverage.
>
> On types: when we say that we did not define types in the formalism, we refer to
> the logical semantics in section "Planning Formalism".  We want to avoid
> many-sorted first-order logic; it would need more space for little
> benefit. There is no loss in generality, as types are just extra predicates in
> the formalism (cf. Helmert AIJ 2009). As mentioned in Footnote 6, we support
> types with the PDDL syntax/semantics.
>
> On contribution: Procedure 1 (our implementation) finds a solution if one
> exists, and for state-bounded tasks, it will find a solution or report when none
> exists. This is the first algorithm with these properties.

---

### Meta-Review · Area_Chair_NsKy · 2024-02-05

**Recommendation:** Accept (Poster)
**Confidence:** 2

**Metareview:**

All reviewers find the topic interesting, and the paper's presentation clear. As such, it would have some value to the planning community. However, it's contribution is somewhat limited: A proof of undecidability nearly identical to that in this paper was published already in 2009 ("Message-Based Web Service Composition, Integrity Constraints and Planning Under Uncertainty: A New Connection", by Hoffmann, Bertoli, Helmert and Pistore, JAIR vol. 35, 2009). The formalization presented in this paper is slightly different, but it does not spell out the relation with previous formalizations. The algorithm for planning with object creation is a basic adaptation of lifted planning, leaving important aspects for future development. In summary, the recommendation is to accept the paper, but this is a borderline recommendation.

Should the paper not be accepted into the main conference, it would make a good contribution to one of the ICAPS workshops, such as HSDIP.

**Ethical Considerations:**

(1) Not Applicable: The paper does not have any ethical considerations to address